# Fine-grained, nonlinear registration of live cell movies reveals spatiotemporal organization of diffuse molecular processes

**Xuexia Jiang**[ID]**, Tadamoto Isogai**[ID]**, Joseph Chi**[ID]**, Gaudenz Danuser**[ID]*

Lyda Hill Department of Bioinformatics, UT Southwestern Medical Center, Dallas, Texas, United States of America

* Gaudenz.Danuser@UTSouthwestern.edu

## Abstract

We present an application of nonlinear image registration to align in microscopy time lapse sequences for every frame the cell outline and interior with the outline and interior of the same cell in a reference frame. The registration relies on a subcellular fiducial marker, a cell motion mask, and a topological regularization that enforces diffeomorphism on the registration without significant loss of granularity. This allows spatiotemporal analysis of extremely noisy and diffuse molecular processes across the entire cell. We validate the registration method for different fiducial markers by measuring the intensity differences between predicted and original time lapse sequences of Actin cytoskeleton images and by uncovering zones of spatially organized GEF- and GTPase signaling dynamics visualized by FRET-based activity biosensors in MDA-MB-231 cells. We then demonstrate applications of the registration method in conjunction with stochastic time-series analysis. We describe distinct zones of locally coherent dynamics of the cytoplasmic protein Profilin in U2OS cells. Further analysis of the Profilin dynamics revealed strong relationships with Actin cytoskeleton reorganization during cell symmetry-breaking and polarization. This study thus provides a framework for extracting information to explore functional interactions between cell morphodynamics, protein distributions, and signaling in cells undergoing continuous shape changes. Matlab code implementing the proposed registration method is available at https://github.com/DanuserLab/Mask-Regularized-Diffeomorphic-Cell-Registration.

## Author summary

By adapting optical flow based, nonlinear image registration we created a method for the time-series analysis of local subcellular processes across the entire cell. This is an extension to our previously published cell edge-based referencing method, which does not allow the extraction of meaningful subcellular time-series more than a few microns away from the cell edge. We leverage the new capacity of sampling at every subcellular location for the discovery of organized Profilin dynamics in symmetry-breaking and polarizing cells, which in turn are related to the dynamics of its regulatory target Actin.

**Data Availability Statement:** The code for cell registration is available at https://github.com/DanuserLab/Mask-Regularized-Diffeomorphic-Cell-

Registration. All raw data is publicly available at https://zenodo.org/record/7351979.

**Funding:** This study was supported by the National Institute of General Medical Sciences (NIH) grant (R35GM136428) and the National Cancer Institute (NIH) grant (R01CA252826) to GD. The funders had no role in study design, data collection and analysis, decision to publish, or preparation of the manuscript.

**Competing interests:** The authors have no competing interests.

## Introduction

Time-series analysis of live cell movies can quantify functional interactions between proteins in the context of complex regulatory networks [1–4]. Utilization of these statistical tools requires extraction of time-series in a cell-centric frame of reference that retains the locations of molecular interactions over time despite cellular movements and shape variation. For cell biology this means following these locations through large morphological deformations such as protrusion and retraction of the cell boundary and associated advective movements of the cytoskeleton and other subcellular structures. Because of the difficulty in tracking subcellular locations over time, many analyses of live cell movies have been limited in their ability to test models of dynamic cell behavior quantitatively.

Historical solutions to the extraction of temporal information in one subcellular location can be broken down into three categories: manual sampling (i.e. kymographs) [5, 6], cell edge propagated sampling [1], and experimental approaches that limit cell deformation [7]. Kymographs, while intuitive and easily used, do not inform on spatiotemporal relations across the entire cell and even locally they tend to cause significant artifacts to the time-series as they generally do not follow cell deformation [5, 6]. The introduction of edge propagated sampling has overcome some of these issues, in principle [1, 8]. However, while registration of locations near the cell edge reveal expected interactions between proteins [1, 2, 9], time-series at locations deeper inside the cell comprise a convolution of the real molecular dynamics with cytoplasmic deformation, leading to uninterpretable data typically a mere few microns away from the cell edge. Experimental constraints such as micropatterning try to overcome this problem by fixing the cell footprint to a particular shape [10], but they do not necessarily limit the advective movements of subcellular structures. More importantly these constraints often introduce harsh perturbations to the cell architecture, which obscure many of the natural cellular behaviors. A more promising approach is the use of image processing to constrain the footprint of the cell across time through nonlinear registration. This approach can either utilize principles of optimal transport or optical flow.

Optimal transport methods find the minimum movement necessary to redistribute the intensity from an input image to the intensity of a target image [11–13]. This is well suited to registration of images collected from multiple samples since minimal displacement is the most appropriate assumption without real displacements between the two images to approximate. Prior work at the cellular scale utilizing optimal transport principles to track time-series in live neutrophils focused on a stereotypical set of behaviors, i.e., the formation of an immunological synapse attacking a model bead [11]. Although in this study the authors compiled a great number of signaling time-series from morphologically diverse neutrophils, the analysis was focused on the narrow synapse region with almost rigid geometry. Neutrophils, whose morphology differed too much from the stereotype, were discarded as outliers. Moreover, the study was designed to analyze relatively coarse cellular processes, which softened the requirement for spatial alignment of the time-series with detailed variations of the synapse shape.

Optical flow-based methods also work under the assumption of intensity conservation but estimate local displacements based on local intensity differences and the observed intensity gradient [14–16]. These methods have been applied to great success in registration of functional brain studies that also seek to extract time-series with high spatial granularity [17]. However, the computation of optical flow belongs to a class of underdetermined inverse problems that requires regularization. Although much of the recent work in optical flow analysis has deployed deep learning approaches to accomplish implicit regularization[18], in many scenarios with priors, for example, on the length scale of variation in the displacement field, it remains advantageous to deploy application-specific constraints to stabilize the flow

reconstruction [19, 20]. Here we present an optical flow-based approach with a regularization scheme tailored for cell biological applications. The method enables the registration of the cell outline and of image patterns within the cell perimeter throughout live movies. We then illustrate how the registration can be employed to extract local time-series across the entire footprint of arbitrarily deforming cells. To demonstrate the potential of this time-series extraction pipeline we chose to analyze FRET-based biosensor activation signals and the dynamics of Profilin, which is a diffuse cytoplasmic protein implicated in the regulation of Actin polymerization. Profilin's interaction with Actin cannot be mapped without cellular registration.

## Results

### Cell-centric frame of reference for the analysis of the spatiotemporal dynamics of diffuse proteins in live cell movies

To extract time-series from live cell movies all frames must be mapped from the lab-centric frame of reference of the image detector into a cell-centric frame of reference, where any subcellular location covers a relevant time-series. This is an ill-defined problem since we cannot directly observe all physical and chemical processes that govern the transport of a labeled protein or its activation over time, including the changes in cell morphology itself. Nevertheless, by using an approximation of these processes using optical flow principles we often can capture subcellular dynamics near the resolution of the imaging.

Among the various techniques for computing optical flow, Thirion's demons have gained popularity because of fast computation speed, intuitive approach, and ease of adjustment of the regularization to diverse data sets [21–23]. Thirion's demons define correspondence between two images following locally the intensity gradient to approximate the advective process that took place between the two images [21]. The procedure thus makes minimal assumptions about the data and performs well when a movie is sampled quickly relative to the changes in morphology, a shared prerequisite for the intended goal of time-series analysis.

Our goal is to facilitate the analysis of subcellular patterns of dynamic molecular activities visualized via fluorescently labeled proteins in 2D movies of single cells. We conceptualize the cell as a dense space where the location of a molecular activity or concentration, referred to as the signal of interest, is coupled to the morphodynamics of the cell (Fig 1a, second row). The morphodynamics is observed by a separate probe, referred to as the location fiducial (Fig 1a, top row). Using the location fiducial, we apply nonlinear image registration techniques with the objective of matching all location fiducial images in the movie as closely as possible to a reference frame (S1 Movie, top). The reference frame can be any image of the same location fiducial, including an image of another cell. Without losing the power of generalizing the pipeline, we will focus here on demonstrating the scenario where the middle frame of a movie is used as the reference frame. We then remap images of the corresponding signal of interest using the deformations that produced closely matched fiducial images to compile a movie of the signal of interest with a fixed cell shape (Fig 1a, third and fourth row). The resulting new movie displays the spatial and temporal dynamics of the signal of interest that were not coupled to the advective movement of location fiducial and cell boundary (S1 Movie, bottom).

The implemented pipeline rests on important assumptions about the data and signal of interest: i) The sampling rate is below the Nyquist limit of the biological behavior of interest. This implies small shape changes between timepoints but large shape changes can occur over the course of the entire movie. ii) The cell should sit flat in 2D and should not move out of view for the duration of the analysis. While the cell can contact other cells, it should not move below or above other cells as in 2D this produces a region of high intensity in the fiducial channel that will be separated by the remapping process. iii) The subcellular motion is accurately

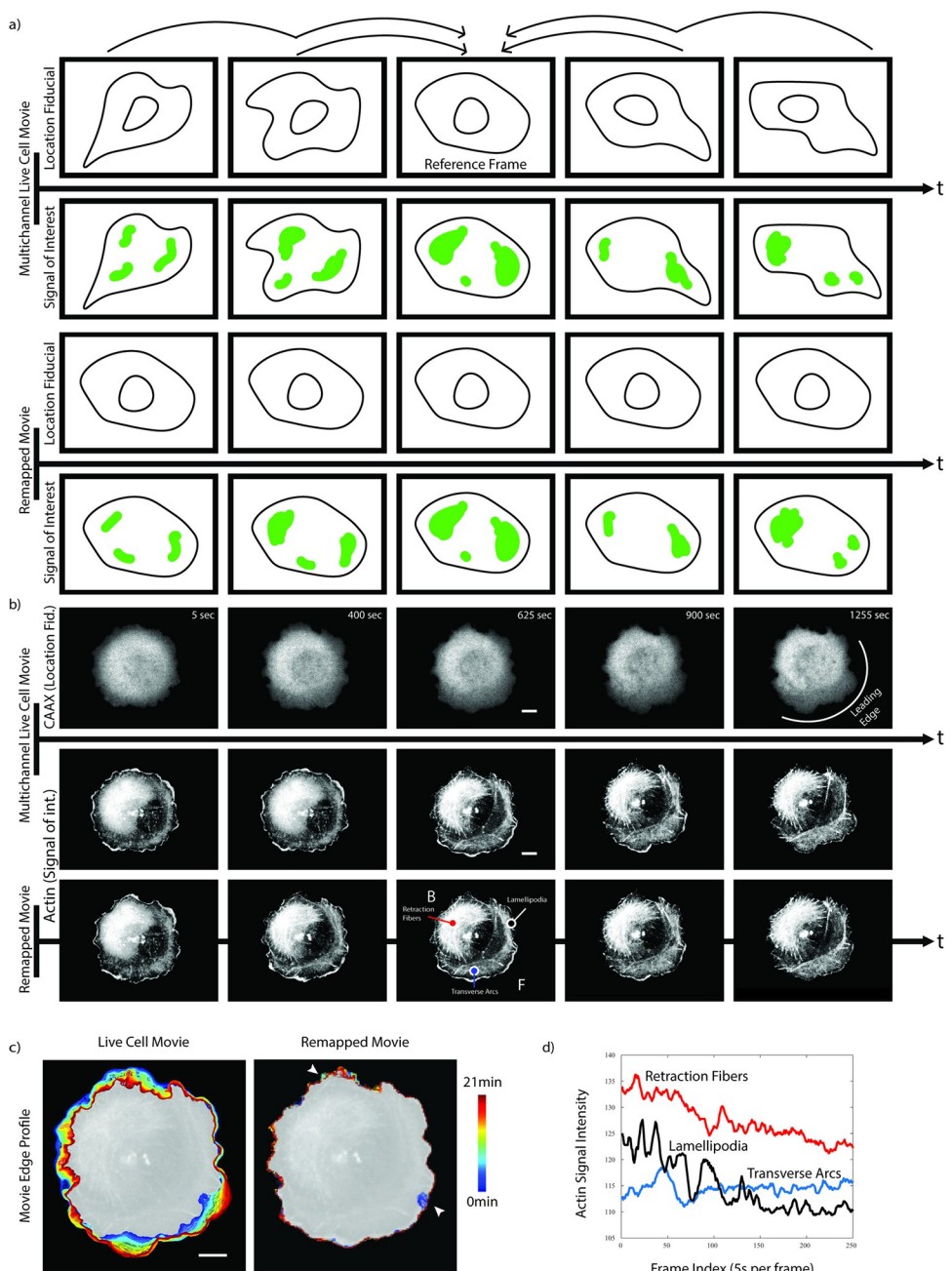

**Fig 1. Registration of a signal of interest captured by a live cell movie using a co-imaged location fiducial.** a)
Illustration of the cell registration pipeline. Top row: The pipeline establishes a frame-by-frame diffeomorphism
between all images of a location fiducial time lapse sequence and a reference frame. Without loss of generality, all
analyses in this paper rely on the center frame of the sequence as the reference frame. Bottom row: The diffeomorphic
functions are applied to remapping a signal of interest from the original movie to a new movie with a rigid geometry
defined by the location fiducial reference. The temporal dynamics of the signal of interest is conserved by the
remapping process. b) Application of the pipeline to a cell undergoing symmetry breaking, i.e. it transforms from a
round geometry to a geometry with front (F)-back (B) polarity (see annotation in the center frame of the remapped
Actin signal). The data includes a Halo-CAAX and mNG-Actin time lapse sequence as a location fiducial and as the
signal of interest, respectively. Colored points in the center frame of the remapped Actin signal indicate the three
positions at which mNG-Actin time-series are sampled in d. c) Cell edge color-coded from blue (early time points) to
red (late time points) in the geometry of the original movie (left) and in the geometry of the reference frame (right).
With the exception of two regions with strong, intermittent ruffling activity (arrowheads), the registration produces a
stationary cell edge. d) The registration of the time lapse image sequence in a spatially stationary reference frame

permits straightforward sampling of time-series of the signal of interest. The traces indicate mNG-Actin signal fluctuations at stereotypical sites of interest: Red–retraction fibers, Blue–transverse arcs, Black–lamellipodia. All scale bars 10 μm.

represented by a location fiducial marker. While this assumption drives the primary objective of cellular registration, our choice of regularization substantially relaxes the requirement for complete tracking of the fiducial marker throughout the cell perimeter.

Fig 1b introduces the pipeline on the example of a polarizing U2OS cell. Over the course of approx. 20 minutes the cell changes from a rounded shape to a canonical migratory shape with a leading edge. We use Actin as the signal of interest and a CAAX membrane marker as the location fiducial. The method eliminates shape deformation except for spurious artifacts in two peripheral regions of erratic ruffle formation where cell edge tracking fails (Fig 1c and S2 Movie).

After remapping, time-series of the signal of interest can be extracted at any subcellular location (Fig 1d, positions of lamellipodia, retraction fiber, and transverse arcs are indicated in the reference time point of Fig 1b 3$^{rd}$ row). Of note, in this example the registration is accomplished based on a diffuse signal associated with the plasma membrane. Nonetheless, fine fibrous structures of the Actin signal of interest are preserved at both the cell front and the back (Fig 1b 3$^{rd}$ row), indicating the accurate estimation of a fine-grained deformation field over time.

## Algorithm for fitting deformation fields

The original Thirion's demons algorithm framed image registration as an advective process in which a moving image (movie frames) is deformed locally onto a target image (the reference frame) such that the remapped moving image, through interpolation, matched the target image as closely as possible [21, 22]. The algorithm alternates between calculating the local displacements based on intensity differences and spatially regularizing the displacements with a Gaussian kernel. For a given coordinate, let $m$ be the local intensity of the moving image and $f$ be the local intensity of the target image. The local displacement $\boldsymbol{u} = [u_1, u_2]$ is given by:

$$\boldsymbol{u} = \frac{(m-f)\nabla f}{\alpha^2(m-f)^2 + |\nabla f|^2} \qquad (1)$$

Here, $\nabla f$ denotes the gradient of the target image intensity and $\alpha$ is an optional throttling term usually set to limit the maximal displacement calculated at each iteration to 1 pixel. A deformation field $\mathbf{U} = [\mathbf{U}_1, \mathbf{U}_2]$ concatenating two matrices containing the first and second displacement components across the image, respectively, is then smoothed by a Gaussian kernel $K_{diff}$ and the entire process repeated $n$ times until a registered image is achieved.

$$\boldsymbol{u} \leftarrow K_{diff} * \boldsymbol{u} \qquad (2)$$

The solution for any given two images depends on the setting of $\alpha$ and $K_{diff}$, which determines the influence of small, local structures on the image matching.

The original algorithm was developed for photography of natural scenes. Compared to this kind of imagery, live cell movies contain higher noise and lower dynamic range. This creates a scenario where the result is highly dependent on $\alpha$ and $K_{\mathbf{diff}}$ values, where low values will cause the algorithm to get stuck in local minima and large values will cause the algorithm to not register small cell protrusions with known biological importance. To solve this, we introduced a mask regularization component, in which the mask is a segmentation of the cell

separating the cell foreground from background and other cells in the field of view:

$$\boldsymbol{u} = \frac{(m - f)\nabla f}{\alpha^2(m - f)^2 + |\nabla f|^2} + \frac{(m_{\text{mask}} - f_{\text{mask}})\nabla f_{\text{mask}}}{\alpha^2(m_{\text{mask}} - f_{\text{mask}})^2 + |\nabla f_{\text{mask}}|^2} \tag{3}$$

Here $m_{\text{mask}}$ is the location of the segmentation mask in the moving image and $f_{\text{mask}}$ is the location of the mask in the target image. We derive the mask gradient $\nabla f_{\text{mask}}$ over the entire cell footprint from the gradient of the map of shortest distances to the cell edge (Fig 2a). In practice the mask regularization component is non-zero where the two image segmentations do not overlap. Thus, it guides the registration process to an overlay of the two cell images moving the registration out of local minima. In presence of a high-quality segmentation of the cell outline, the regularization component is independent of the noise of the location fiducial. This permits us to set $K_{\text{diff}}$ based on the expected diffusion distance between time steps as opposed to an arbitrary number for better registrations. We then set $\alpha$ to limit displacements to 1 pixel, but higher limits can be used to quickly test suitability of input data for this pipeline (not shown).

The original publication of Thirion's demons presented a way to enforce diffeomorphism based on Lie group theory [21]. The approach broke each displacement into a series of smaller steps based on the magnitude of the displacement and smoothed for each of these steps. This approach tends to be trapped in local minima, especially with a parameter selection meant to capture the granularity in cell data [23]. We chose instead to constrain the mapping to a diffeomorphism via an explicit topological sorting of the displacements. Given the matrix of unsorted displacements $\mathbf{U}$, we compute $\mathbf{U}_{\text{sorted}} = [\mathbf{U}_{1, \text{ sorted}}, \mathbf{U}_{2, \text{ sorted}}]$, with

$$\mathbf{U}_1 = \begin{bmatrix} u_1(1, 1) & \cdots & u_1(1, n) \\ \vdots & \ddots & \vdots \\ u_1(m, 1) & \cdots & u_1(m, n) \end{bmatrix} \text{ and } \mathbf{U}_2 = \begin{bmatrix} u_2(1, 1) & \cdots & u_2(1, n) \\ \vdots & \ddots & \vdots \\ u_2(m, 1) & \cdots & u_2(m, n) \end{bmatrix},$$

such that

$$\forall \mathbf{U}_1 \in \mathbf{U}_{1,\text{sorted}} : \ \mathbf{U}_1(., a) + a \geq \mathbf{U}_1(., b) + b \qquad \forall a > b \tag{4$'$}$$

$$\forall \mathbf{U}_2 \in \mathbf{U}_{2,\text{sorted}} : \ \mathbf{U}_2(a, .) + a \geq \mathbf{U}_2(b, .) + b \qquad \forall a > b \tag{4$''$}$$

To illustrate this diffeomorphism constraint, a deformation field $\mathbf{U}$ is indexed as a mesh for interpolation (Fig 2b, panels i and ii). Square meshes indicate an identity transform and non-square meshes indicate local image deformation. A break in diffeomorphism occurs when mesh indices are out of order, leading to a crossing of edges (Fig 2b, panel iii). Per iteration the sorting procedure corrects violations of the diffeomorphism (Fig 2b, panel iv) by reordering the mesh indices. In practice, diffeomorphism breaks occur when either a fine-grained pattern or noise push the optimization towards a registration with crossings. With our sorting implemented, such displacements will be suppressed.

To illustrate the performance of the proposed cell registration we chose a U2OS cell undergoing a stereotypical isotropic spreading process and remapped two distant time points (160 frames sampled at 5s per frame) while setting $K_{\text{diff}}$ to a low value of 0.5 pixels. This condition deviates from our expectation of fast sampled changes but allows easy interpretation of the impact of our modification to the original Thirion's demons and shows that our approach can handle more extreme morphology changes. In combination, the motion mask and sorting-based regularization permit near-pixel perfect edge alignment between the two distant time points (Fig 2c). Without these constraints, a typical remapping can exhibit perimeter-normalized boundary deviations (total pixel difference in image masks/length of

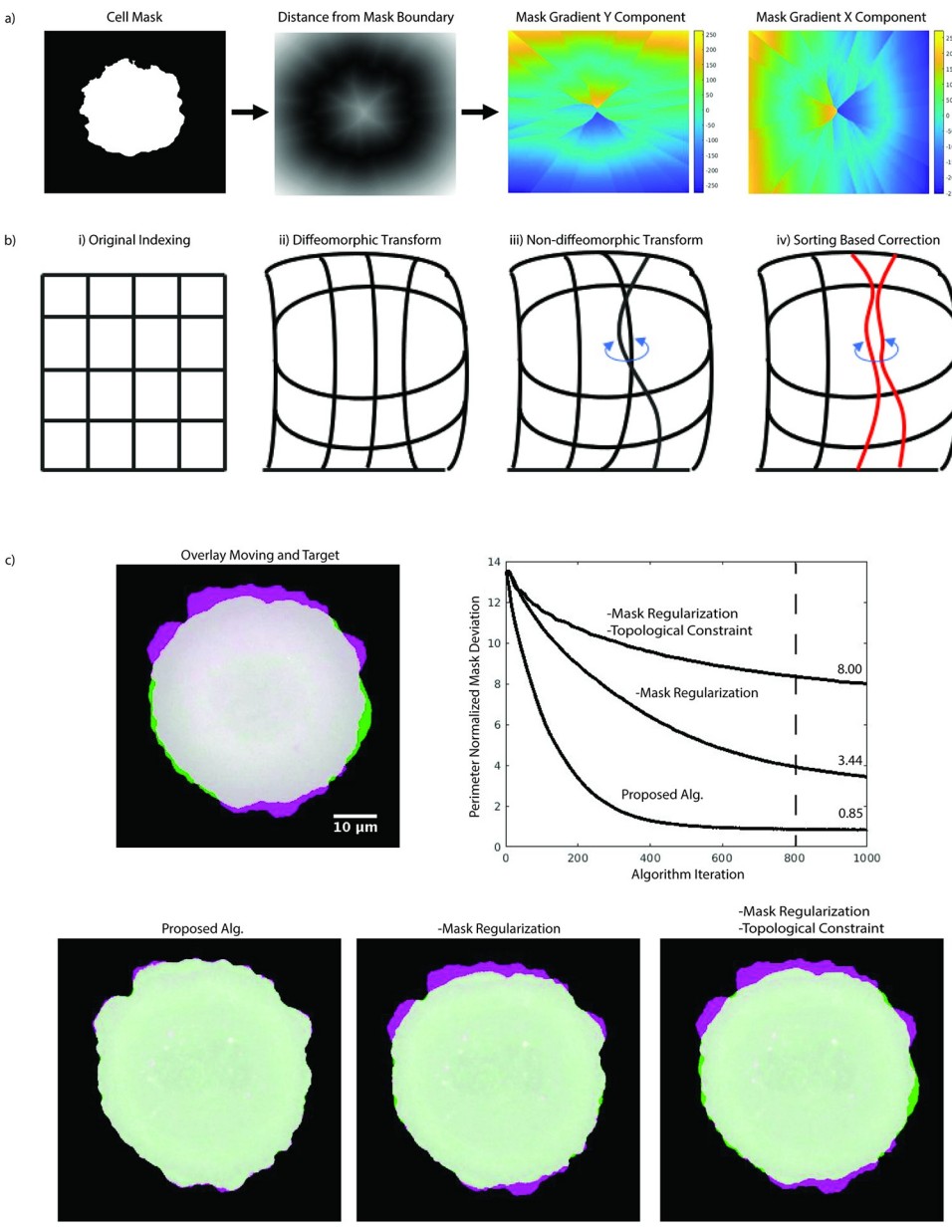

**Fig 2. Computational elements for the remapping process.** a) Extraction of the mask gradient $\nabla f_{\text{mask}}$. Based on a cell mask, we calculate for each pixel the distance to the nearest cell boundary element for both the cell-interior and cell-exterior spaces. The gradient on this distance field defines $\nabla f_{\text{mask}}$. b) Diffeomorphism constraint. From left to right, illustrations of interpolation fields, where edges indicate the sampling position of an input image to remap onto a target. The mesh diagram shows displacements as deviation from a square mesh (1st diagram from left). A diffeomorphic transform (2nd diagram) is represented by a deformed mesh. A break in diffeomorphism (3rd diagram) is represented by crossings of mesh edges. By sorting the mesh coordinates in sequential order, breaks in diffeomorphism can be reverted (4th diagram). c) Effect of algorithm components on cell edge registration. Accuracy of registration over $n$ iterations is indicated by the area of mismatch (green and purple) between the moving cell and the target cell image, normalized by the target cell perimeter. Removing the mask regularization and topological constraint enforcing diffeomorphism reduces both the rate of convergence and the final accuracy. The dashed line indicates the iteration stop for a visualization of the registration results (bottom row). The proposed algorithm gives a near pixel perfect registration of the two images. Removing the mask regularization largely reduces the rate of convergence. Removing both mask regularization and topological constraint causes failure in the capture of small protrusions.

perimeter in pixels) larger than 8 pixels, which means a loss of an entire protrusion event. Our modified algorithm additionally requires fewer iterations to converge to an accurate edge alignment, showing that accuracy is not traded with computational inefficiency.

We empirically determined the iteration number $n = 200$ as sufficient for all our movies to achieve near-pixel alignment for the cell mask (Fig 2c). This value is fixed in the current software distribution, but can be readily adjusted by the user. To remap a sequence of moving frames onto a final reference frame, we change the target image after $n$ iterations to the next frame in the sequence (Fig 1a). Thus, frames at time points farther from the reference frame receive more passes through the algorithm.

## Evaluation of algorithm by image signal remapping

Our approach relies on a subcellular location fiducial. Generally, this requires an additional fluorescence channel for live cell imaging besides the signal of interest. To test the performance in cellular registration for location fiducials with different spatial characteristics we acquired three channel movies of U2OS cells labeled with mNeonGreen (mNG)-Actin, SNAP-Profilin and Halo-Vasp. For a visual test we first overlaid two mNG-Actin frames of the movie separated by 8 minutes in green and magenta pseudo-colors (Fig 3a). Whereas in the total Actin signal differences between the two frames were discernible only at the cell periphery, they became more obvious for Actin fibers in the cell center in an overlay of the two frames containing only the high intensity components. We thus tested registration performance for both the total Actin signal and the high-intensity signal as the signals of interest. To do so we used the SNAP-Profilin and Halo-Vasp channels as the location fiducials (Fig 3b). Profilin is a cytoplasmic binding partner of monomeric Actin and presents a largely diffuse signal. In concert with Actin polymerases, including formins and Vasp, it serves as a pacemaker for Actin filament elongation [24–26]. Vasp localizes at the tip of polymerizing Actin filaments and thus is visible as a punctate cytoplasmic signal colocalized with focal adhesion complexes with a high concentration of terminating Actin filaments [27].

The use of a dedicated fluorescence channel only for registration can be quite cumbersome and can increase photo-toxicity. Since the goal of the cell registration is the extraction of informative time-series, we can treat the signal of interest as a mixture of high spatial frequency signals describing local molecular activity and low spatial frequency signals describing the coarse scale dynamics of subcellular molecular organization. Assuming separability of the spatial frequencies, we can use the low frequency bands for registration without artificially flattening the informative high frequency signals. In the example of Fig 3b, we therefore computed as location fiducial a lowpass-filtered version of the mNG-Actin signal by applying a Gaussian filter with $\sigma = 20$ pixels.

Real world photography and medical image registration use landmark comparison and reconstruction of computationally distorted images to measure performance. Neither is a feasible metric for our live cell movies as we have no ground truth. Moreover, a simulated distortion can arbitrarily favor a particular fiducial choice. We therefore introduce the 'to-target transform' and the 'half-distance transform' accuracies as alternative performance measures for the remapping quality (Fig 3c). For both measures, we first remapped the signal of interest from the moving image to the target image using location fiducial deformation fields over two frames. To ensure that broadly visible movement exists between consecutive time points we tenfold down-sampled the original movie (50 s frame interval instead of the original 5 s interval). For the to-target transform accuracy we then asked how similar the remapped signal of interest is in comparison to the imaged signal of interest in the target image. The proximity between remapped and target signals of interest was determined by the average pixelwise

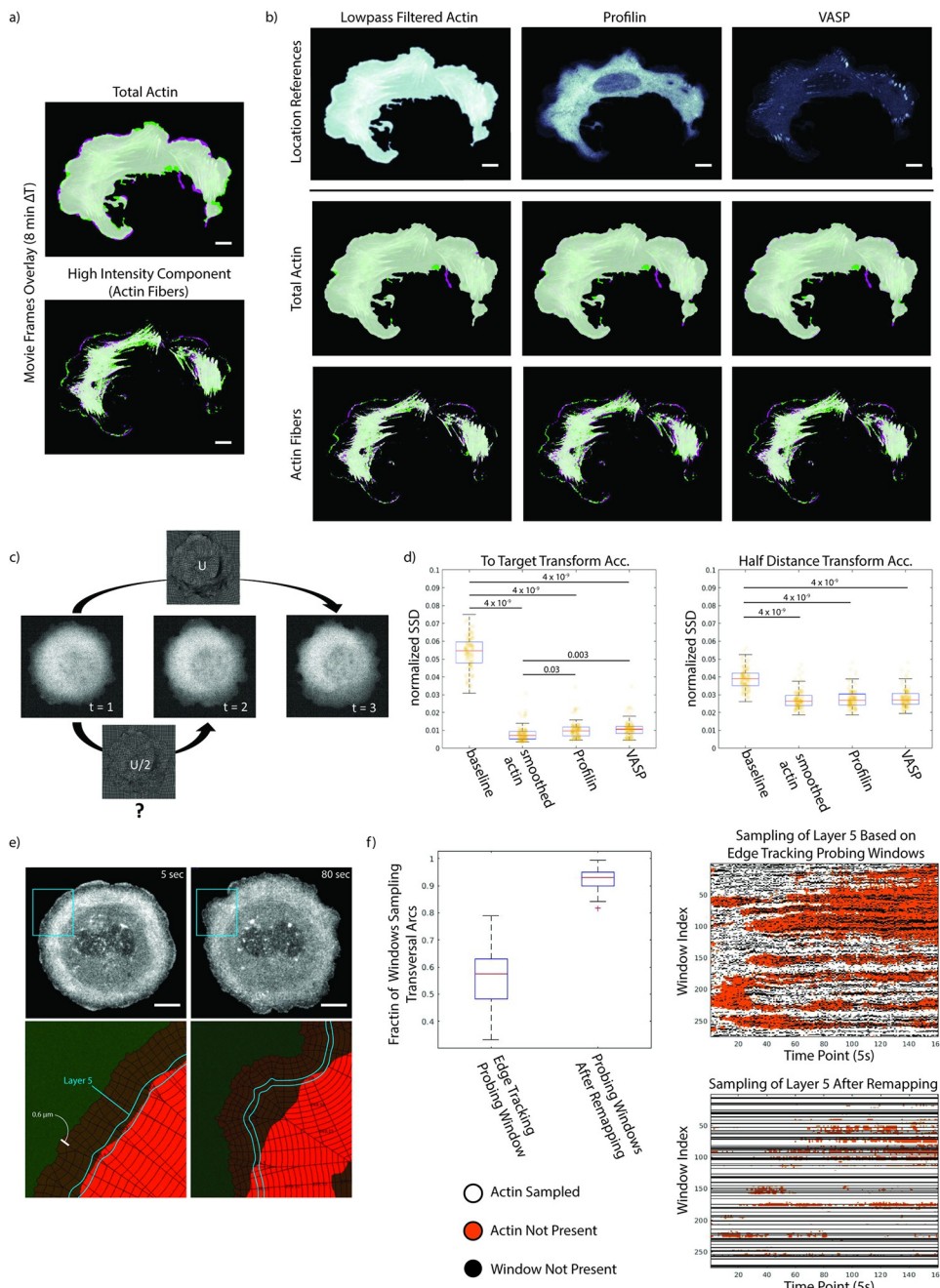

**Fig 3. Impact of location fiducial on remapping accuracy.** a) Overlay of two frames separated by 8 minutes of the total Actin channel (top) and a high intensity component (defined by manually selected intensity cutoff) emphasizing Actin fibers (bottom). b) Qualitative comparison of the remapping quality for both Actin signals of interest using the lowpass-filtered Actin image (top row left), Profilin (top row middle), and Vasp as location fiducials. There were minimal differences in the registration quality in both the total Actin (middle row) and Actin fiber (bottom row) components of the Actin signal of interest. c) Definition of the half-distance transform used as a ground truth for performance evaluation. First, we estimate the diffeomorphic map between moving frame (t = 1) and target frame (t = 3). We then asked how well a half diffeomorphic map matches the middle frame (t = 2) under the expectation that the mapping between moving and target frames follows a linear deformation path. d) Quantification of subcellular remapping accuracy (sum of squared distance (SSD) between target and remapped images) using the to-target and half-distance transforms for indicated location fiducials. The SSD between untransformed moving and target frames is computed as a baseline. Box plots illustrate 25th, 50th, and 75th percentile of n = 116 moving/target frame pairs pooled from m = 4 movies. Whiskers indicate the 5th and 95th percentile. P-values are calculated by one-way ANOVA testing. Overall, there were only small differences in accuracy between the choices of location fiducials. e, f) Comparison of

time-series sampling using a fixed grid of probing windows in the reference frame of a full cell registered movie vs. using edge-tracking probing windows. e) Snapshots of the actomyosin organization in a U2OS cell at an early (top left) and later (top right) stage before symmetry breaking. The cell displays characteristic transversal arcs behind a thin lamellipodia layer at the periphery. Towards the symmetry breaking event, the arcs begin to dissolve. In between, the transversal arcs follow the displacement of the cell edge. We detected these arcs using an intensity filter (red regions bottom row). Overlaid to the segmented arcs region is a grid of 0.6 μm wide (at initial time point) and 0.6 μm deep edge-tracking probing windows. Highlighted in blue, 5th layer of probing windows, which sample the transverse arcs in early time points (left). In late time points the windows fall outside the arc region. f) Sampling of the transverse arcs by the 5th layer of probing windows that are either tracking the cell edge in the original movie or fixed after remapping the movie to a reference frame (left). Space vs time heatmaps of the samples (right, top & bottom). The maps show the presence (white) or absence (orange) of transverse arcs. Edge-tracking windows have frequent "drop-outs" (black) in layers further away from the cell periphery. Window "drop-outs" (black) are persistent in time when the movie is remapped. All scale bars 10 μm.

squared intensity difference. This metric quantifies how much of the dynamics is captured by the location fiducial and the remapping process. For the half-distance transform accuracy, we divided the magnitude of the deformation field over two frame intervals in half and compared the thus remapped signal of interest to the signal of interest in the skipped frame (Fig 3c). This metric quantifies how well the approximations of an advective process underlying the remapping capture the real spatial dynamics of the signal of interest.

Equipped with these two performance measures we remapped the raw as well as high-intensity Actin structures as the signal of interest, using Profilin, Vasp, and lowpass-filtered Actin as location fiducials. We included the sum of squared intensity differences between unregistered original images over two frames as a baseline (background set to 0). Like in Fig 1, we chose Actin as the test case because of the dynamic and multi-factorially regulated structure, making it unlikely that any of the fiducials would fully capture the evolution over time. Overall, the half-distance transform accuracy is worse than the to-target transform accuracy (Fig 3d). This is mostly attributable to poor registration near the cell edge, which moves on a much faster timescale than the cell interior [1, 28]. Both Profilin and Vasp as location fiducials greatly outperformed the baseline deviation, despite the diffuse image character in the former and the relatively scarce punctate pattern in the latter. Similar performance was accomplished in cells that expressed a diffuse Halo-CAAX membrane marker as a location fiducial rather than fluorescent Vasp (S1 Fig). Unsurprisingly, in both tests the highest remapping accuracy was achieved with lowpass-filtered Actin as the location fiducial (Figs 3d and S1B). This shows that for generating precisely stabilized images of cytoskeleton structure in a reference frame our algorithm works best using a low pass filtered copy of the original signal for alignment.

## Evaluation of the remapping algorithm for the extraction of subcellular time-series

To further establish confidence that the remapping algorithm permits extraction of meaningful time-series throughout the cell and over the entire duration of a movie, we compared the performance against our well-established cell edge-tracking windowing method [1, 8]. In brief, this method tracks the cell edge and places a grid of probing windows indexed by radial position and depth in layers. While the newly proposed registration-based approach can handle a variety of cell behaviors, the windowing strategy has largely been used in scenarios of a relatively stationary edge dynamics. We, therefore, chose a less motile U2OS cell (labeled with mNG-Actin, Halo-CAAX, SNAP-Profilin) undergoing stereotypical cell spreading to compare the two methods.

In U2OS cells the periphery is demarcated by distinct and persistent network of transversal Actin arcs. We segmented the arcs throughout the movie using a simple intensity threshold

and hole-filling operation (Fig 3e). We then applied the cell edge-tracking windowing approach to define layers of 0.6 μm deep probing windows that follow the edge movement (S3 Movie). In early time points the arcs begin ~3 μm away from the cell edge (sampled by layers 5 to 8). We chose to examine the sampling of the circumferential Actin region in layer 5 in a movie where the Actin signal of interest was first remapped based on the CAAX membrane marker as location fiducial. In the remapped movie we applied the window positions from the reference frame to the entire movie as a stationary probing grid. Due to the fixed one-to-one window correspondence between layers, deeper layers of the edge-tracking probing windows exhibit drop-out events (Fig 3f black streaks on the space-time representation of layer 5). Importantly, these drop-out windows do not persist over time since they are dependent on the geometry of the cell edge. In contrast, the window grid in a remapped movie has persistent drop-outs. The band between cell edge and transversal arcs, i.e., the lamellipodium, varies in width over time and also in space. Accordingly, in a sampling approach that preserves a constant distance from the edge, windows at the transition between lamellipodium and arcs alternate in the structure they sample. In contrast, the proposed cell registration accounts for the variation in subcellular structures. Indeed, while edge-tracking probing windows in layer 5 sampled the transversal arc structure in only 58% of the windows and time points, 92% of the stationary windows of layer 5 in the remapped movie sampled transversal arcs (Fig 3f). The loss of connection to subcellular structures has been a serious limitation for many studies relying on edge tracking probing windows as they sample time-series associated with distinct regulatory regimes [29, 30]. The proposed remapping resolves this issue now for subcellular structures that follow the diffeomorphism defined by the location fiducial.

## Spatial coherence in fluctuation time-series of FRET-based biosensors of molecular signaling activity increases with cell registration

Using edge-tracking probing windows for the local sampling of time-series in a cell edge-centric frame of reference we have in the past conducted studies of the coupling between cytoskeleton and signaling dynamics and the movement of the cell edge [1, 31–34]. As indicated in Fig 3, this approach is limited to a relatively narrow band along the cell edge and it makes the strong assumption that the larger organization of the signal of interest follows the edge movement. With the proposed cell registration we may be able to relax this condition as we remap the signal of interest under the influence of a fiducial marker throughout the entire cell. To test the effect of the cell registration on capturing the subcellular organization of a dynamic signal of interest, we turned to previously published and analyzed live cell movies of GTPase signaling activity and their upstream regulators visualized by Foerster Resonance Energy Transfer (FRET)-based biosensors [34]. Specifically, we examined the alignment of movies displaying the activation of the GTPase Cdc42 and its activating Guanine Exchange Factor (GEF) Asef (Fig 4a). Compared to the Actin cytoskeleton images in Fig 3, the activity of these molecular signals is more diffuse, albeit visibly organized in space. We registered the movies to the middle frame as the reference (Fig 4b), using the donor fluorescence of the Cdc42 FRET-sensor as the location fiducial. The pattern of the donor images is dominated by the topographic variation of the cell. Therefore, the location fiducial controls in this case the remapping of the signal of interest primarily for intensity shifts associated with changes in cell thickness that arise with cell movement.

   To test whether the registration yields a gain in the spatial information that is contained by a diffuse signal of interest we created an operator that computes the local spatial coherence of the signal dynamics. The operator considers the time-series sampled in fixed pixels or superpixels in a 3x3 neighborhood and determines the mean correlation of the 36 unique time-series

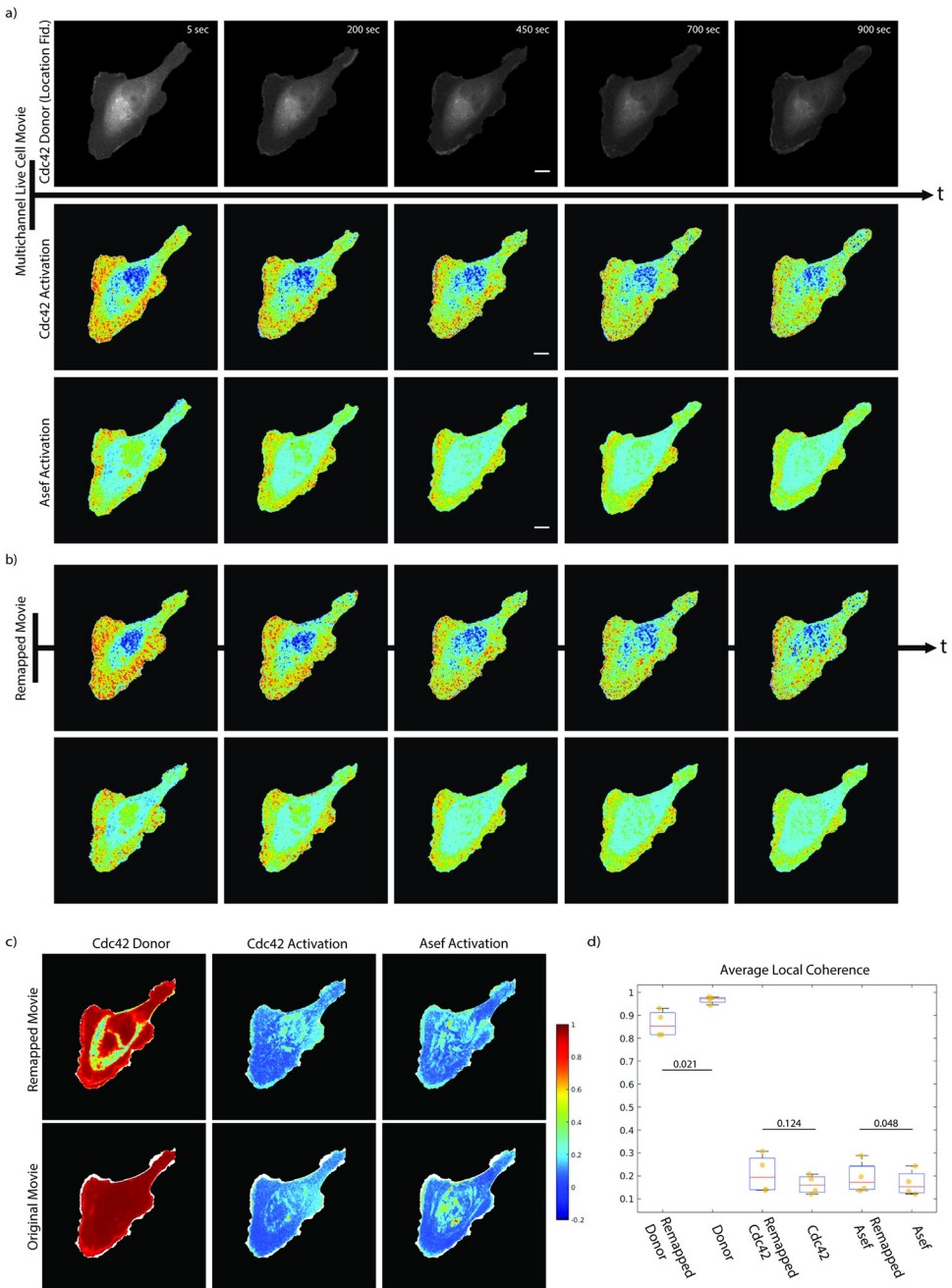

**Fig 4. Analysis of spatial coherence in diffuse molecular signals.** a) Select frames of a movie visualizing the activation of the GTPase Cdc42 (middle) and its upstream GEF Asef (bottom) using multiplex FRET-based biosensors. Shown are normalized FRET-ratio signal indicating local activation of the molecues. For registration the donor signal of the Cdc42 biosensor is used as the location fiducial (top). b) Remapped, normalized FRET-ratio signals for the same time points as in a). c) Spatial coherence of the Cdc42 donor (left), Cdc42 FRET-ratio (center), and Asef FRET-ratio (right) time-series extracted from the remapped (top) and raw time-lapse sequences. d) Distributions of average spatial coherence of the Cdc42 donor, Cdc42 FRET-ratio, and Asef FRET-ratio time-series. Box plots illustrate 25th, 50th, and 75th percentile of average values in for m = 4 movies. Whiskers indicate the 5th and 95th percentile. P-values are calculated by one-way ANOVA testing. Scale bars 10 μm.

pairs among the 9 time-series. The coherence value falls between 0 and 1, where higher values indicate self-similarity within the neighborhood. This analysis resembles fluorescence correlation spectroscopy, which typically captures fluctuations at the millisecond time scale [35]. However, at the time scales of seconds, the coherence does not relate to a single molecule property but captures regulatory processes that maintain spatial similarity of fluctuation time-series through either biochemical interactions or local scaffolds. Without spatial registration of the signal of interest to a reference frame the self-similarity of adjacent time-series cannot be detected. Indeed, we observed a systematic increase in coherence for both the Cdc42 and the Asef activity signals when the time-series were sampled in image sequences registered to a reference frame rather than in the original lab frame of reference (Fig 4c and 4d). The increase was relatively mild when considering the entire cell footprint. It was in fact not statistically significant in the case of Cdc42. The reasons for this result are the relatively high noise in the FRET time-series, also reflected by the rather low correlation values in the range 0.2–0.3, and the concentration of the coherence increase in a fairly narrow band along the cell edge where GEF-GTPase interactions are known to be actively regulated [1, 36] (Fig 4c). In contrast, after registration the coherence in the FRET-donor signal significantly decreased from a value close to 1 for pixels sampled in the original lab frame of reference (Fig 4c and 4d). The decrease in coherence is concentrated in the perinuclear region, where the improved spatial alignment after cell registration reveals locally incoherent, volumetric fluctuations in biosensor concentration that are blurred by cell motion in the lab frame of reference. In summary, these experiments indicate how the registration of the cell footprint over a time lapse sequence permits the recovery of fine-grained spatial patterns in the temporal fluctuation of otherwise diffuse image signals.

## Time-series of Profilin reveal spatiotemporal organization of dynamic concentrations

Equipped with tools for sampling and unveiling fine-grained spatial patterns in the temporal fluctuation of otherwise diffuse signals across an entire cell, we turned to examining the subcellular organization of Profilin. Profilin is a 15kDa molecule binding monomeric Actin and Profilin-Actin is considered the physiological substrate of filament growth [25, 26, 37]. Numerous biochemical experiments have shown specific interactions between Profilin-Actin and key cytoskeletal regulators (such as formins) suggesting that distinct Actin structures would colocalize with local pools of Profilin with distinct dynamics [24, 38]. Endogenously SNAP-tagged Profilin in live cells displayed a diffuse signal with no visually discernable pattern beyond intensity variations that related to the integration along the optical axis of fluorescence in variably thick cells (Fig 5a–5c, left).

We anticipated that local time-series coherence analysis would uncover a hidden spatial organization in the Profilin dynamics, akin to the GTPase- and GEF-signals. We performed this analysis on 3 cell populations: First, U2OS cells expressing cytoplasmic mCherry as a volume marker in a background of mNG-Actin and Halo-CAAX (Fig 5a). Second, U2OS cells expressing fluorescent endogenously SNAP-tagged Profilin, mNG-Actin, and Halo-CAAX (Fig 5b and S4 Movie). These cells were further treated with the myosin-II inhibitor Blebbistatin to induce symmetry breaking throughout the movie, as described by Lomakin et al [39]. This allowed us to follow a putative reorganization in Profilin dynamics in response to a change in global cell morphology. Third, we exogenously co-expressed wild-type EGFP-Profilin (WT) and a mutant mApple-Profilin (R88E) at very low levels using a leaky CMV100 alongside SNAP-Actin in Profilin KO cells (Fig 5c). The R88E mutation abrogates binding to Actin [38]. Therefore, we hypothesized that the dynamics of the mutant Profilin would follow

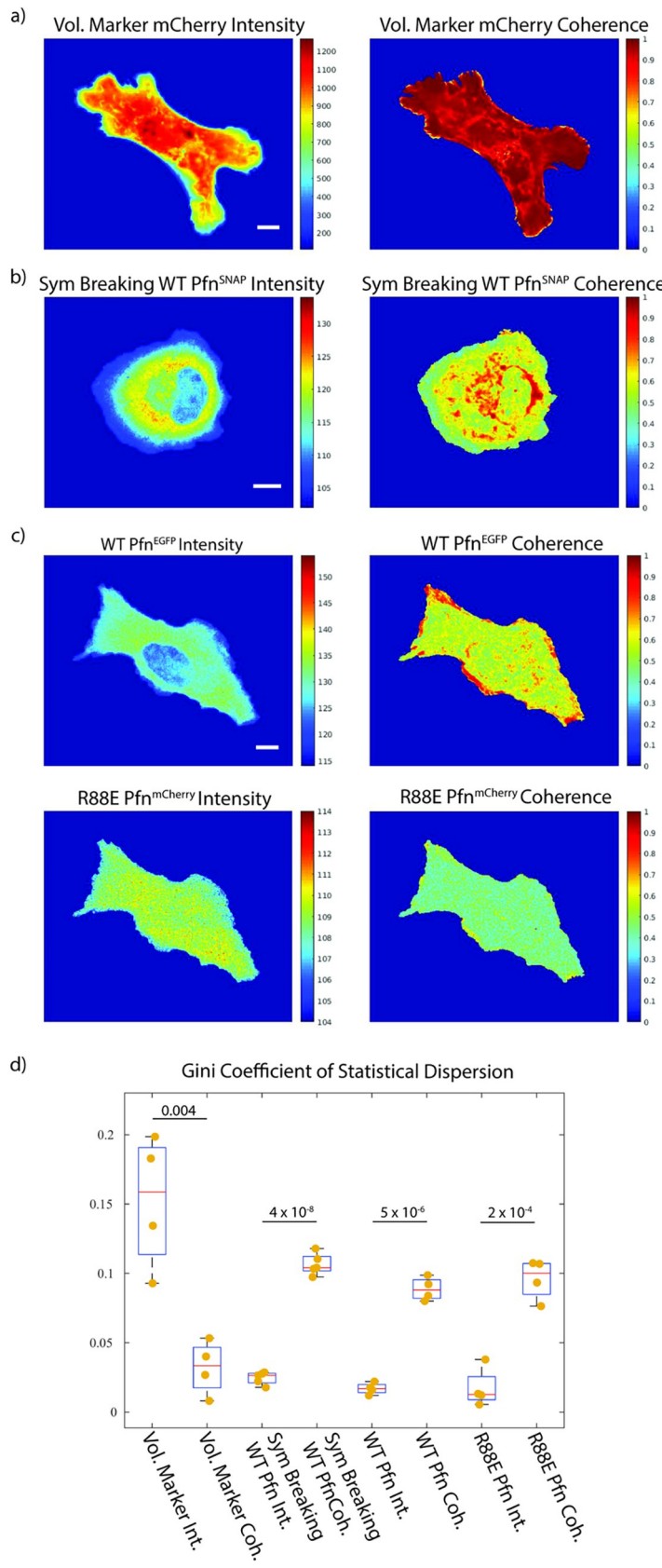

**Fig 5. Profilin dynamics in live cells show patterns in local time-series coherence.** Raw intensity vs local time-series coherence scores of diverse Profilin probes. The computation of reliable coherence scores is enabled by the remapping of the movie into a spatially stationary reference frame. a) Control experiment using an mCherry cytoplasmic volume marker. As expected, coherence of a diffuse signal is high across the entire cell footprint. For remapping, a Halo-CAAX tag membrane marker was used as the location fiducial. b) Cell undergoing drug-induced symmetry breaking, which expresses SNAP-Profilin at endogenous levels. For remapping, a Halo-CAAX tag membrane marker was used as the location fiducial. c) WT EGFP-Profilin (top) and R88E mApple-Profilin mutant (bottom) concurrently expressed at low concentration from a leaky CMV100 promotor in a Profilin knockout cell. For remapping, the lowpass-filtered Actin-SNAP signal was used as the location fiducial. WT and R88E Profilin display distinct coherence patterns. d) Quantification of subcellular heterogeneity of the signals observed in a-c using the Gini coefficient. Dots indicate values for individual cells. All scale bars 10 μm.

a different coherence pattern than that of the wildtype (S5 Movie). In the first 2 cases, we remapped the movies on a central reference frame using the CAAX marker as a location fiducial (images not shown). Due to the limitations in expressing yet another tag for live cell imaging, we remapped the double Profilin labeled cells using a lowpass-filtered Actin channel set to mimic the results of CAAX-based remapping.

The coherence analysis revealed drastically different patterns relative to the raw Profilin signal. High coherence is observed in select sites along the cell edge, in puncta throughout the cytoplasm, and around the nucleus in both the endogenously and exogenously tagged Profilin. We quantified the increase in heterogeneity produced by application of the coherence operator via the Gini coefficient of statistical dispersion (Fig 5d). The Gini coefficient occupies a value range between 0 to 1, where 1 indicates a signal concentrated in one pixel of the cell and 0 indicates a homogeneous signal across all pixels of the cell [40]. For all three cell conditions, the coherence of the Profilin intensity is more heterogenous than the raw Profilin signal, indicating a high degree of local dynamic organization of Profilin, potentially related to its roles in facilitating Actin polymerization. As a control, we performed the same analysis for a cell expressing mCherry as a volume marker. In this case the intensity shows a higher level of heterogeneity than the coherence value, because of significant variation in cell thickness. The coherence of the mCherry volume marker was homogeneously high throughout the cell.

## Profilin coherence is related to actin dynamics

Since WT and R88E Profilin displayed distinct coherence patterns in the same cell, we hypothesized that WT Profilin coherence would show a stronger relationship with Actin dynamics (Fig 6a). Our coherence calculation thus far relied on time-series spanning the entire movie. To test whether Profilin coherence changes in concert with Actin dynamics we computed a coherence sequence using a moving window of 25 frames, i.e. 1/10 of the length of our shortest movies (S5 Movie). The resulting sequence could then be locally correlated with a measure of the Actin signal fluctuations. To match the time scales of changes in Profilin coherence and Actin dynamics, we computed the Shannon information entropy of the Actin signal at every location of the cell using the same moving window approach (Fig 6b and S6 Movie). This signal transform extracts from the overall fairly static Actin images second-to-minute scale processes such as stress fiber growth and movements and retrograde flow.

We then examined the cross correlation of Profilin coherence and Actin entropy near the cell edge (edge– 1.2 μm), around the circumferential Actin network (3.6–6 μm), and around the nucleus (8.4–12 μm) since the data in Fig 5c indicated patterns of high Profilin coherence in these positions. As expected, near the cell edge we observed higher cross correlation between Actin entropy and WT Profilin when compared with the non-interacting R88E Profilin mutant (Fig 6e boxed). The difference is less substantial in deeper regions of the cell because both Actin entropy and Profilin coherence are temporally less salient. This is

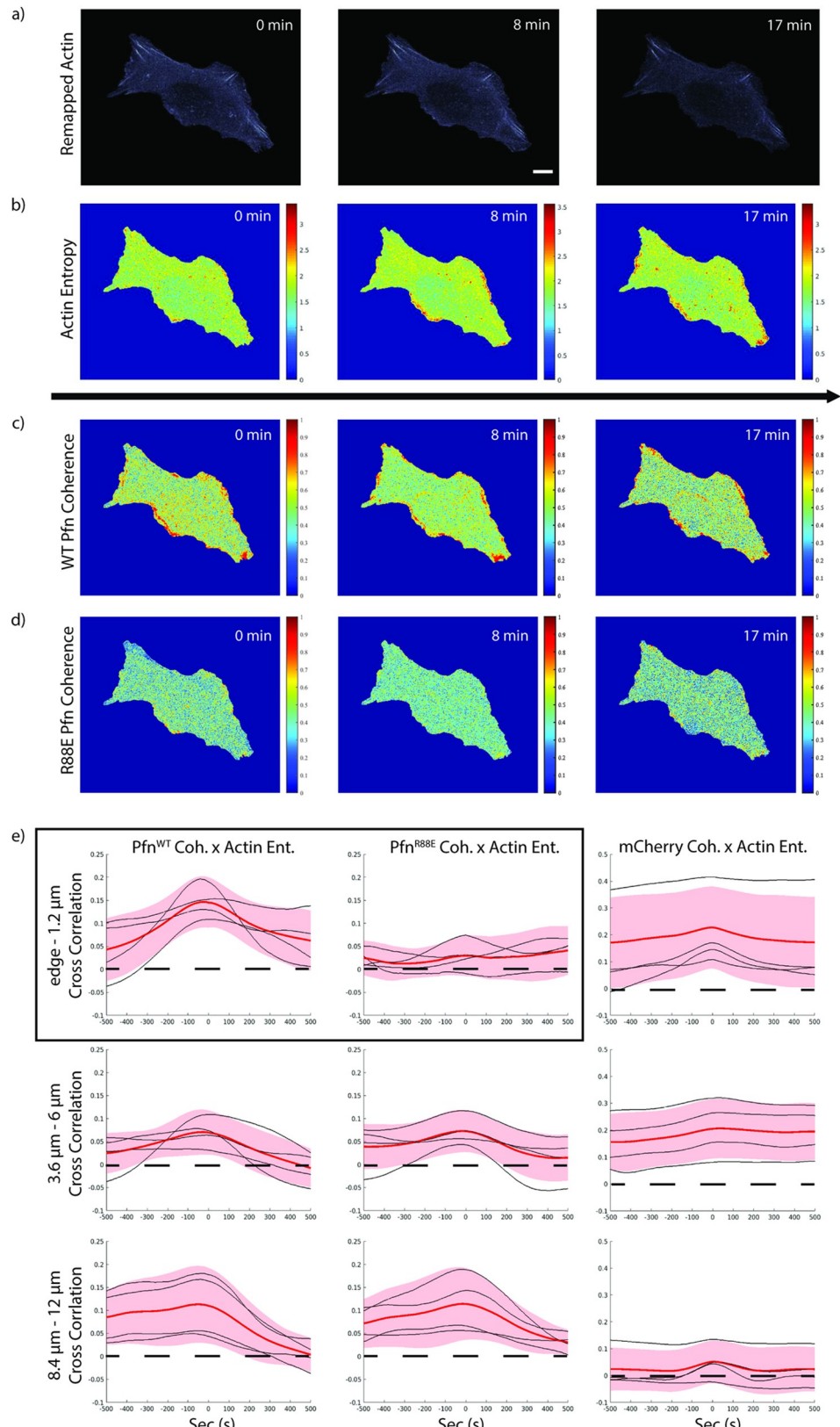

**Fig 6. Co-expressed wildtype and mutant Profilin exhibit different relationship to Actin dynamics.** a-d) Signal transforms to relate Profilin organization to Actin dynamics. The transforms are enabled by remapping the movie into

a spatially stationary reference frame. U2OS cells expressing a) Actin-SNAP with b) entropy of the Actin signal, c) WT EGFP-Profilin, and d) R88E mApple-Profilin. The remapping was accomplished using lowpass-filtered Actin as a location fiducial. To reduce perturbation by overexpression, the two Profilin constructs were expressed in a Profilin null background. Panels in (c) and (d) display the dynamics of the diffuse image signals as local time-series coherence scores over a rolling window of 25 time points (250 s). The SNAP-Actin signal consists of various filament forms as well as a diffuse background of monomers. The mixture makes a direct cross correlation to other molecular processes difficult to interpret. We therefore extracted the entropy over a rolling window of 25 time points, which indicates relative stability of Actin structures (high entropy delineates regions of high polymer turnover and/or concentration variation). e) Cross correlation of Actin entropy and Profilin coherence in subcellular regions defined by distance from the cell edge. We performed this analysis on 3 zones: 0–1.2 μm from the cell edge (top row) roughly corresponding to the thin lamellipodia, 3.6–6 μm from the cell edge (mid row) roughly corresponding to transverse arcs, and 8.4–12 μm corresponding to perinuclear regions. As a control we also calculated the cross correlation between Actin entropy and the coherence of an mCherry volume marker (right column). WT Profilin and Actin show a significant correlation only in the band 0–1.2 μm from the edge, confirming the biochemical function of Profilin as a promoter of Actin polymerization. The correlation collapses for R88E Profilin, which is deficient in Actin binding (boxed). The coupling between Profilin and Actin dynamics is absent in regions more distal from the cell edge. All scale bars 10 μm.

reinforced by the fact that the volume marker's cross correlation with Actin entropy is high in the periphery but low near the nucleus where there is a persistently high volume marker coherence but rare high Actin entropy events (Fig 6e). Together, these analyses indicate how the proposed remapping of signals to a stationary reference frame permits the application of time-series preprocessing in order to extract subtle but significant dynamic behaviors in otherwise diffuse signals for the purpose of assessing the local functional interactions between molecular processes.

## Profilin coherence correlation with actin dynamics is responsive to perturbation

To test the hypothesis that the observed relationship between Profilin coherence and Actin entropy relates to Profilin's functions as a modulator of Actin polymerization, we analyzed cells undergoing symmetry breaking (S7 Movie). Symmetry breaking is a process where the cell transitions from a stable rounded state to a polarized migratory state. In doing so the cell must greatly reorganize its Actin network. After remapping the entire movie to a common reference frame just before the symmetry breaking event, we split the movies into a phase of high morphodynamic activity during symmetry breaking and a phase of slower, steady state morphodynamics after symmetry breaking (Fig 7a). We then computed Actin entropy (Fig 7b) and Profilin coherence (Fig 7c) for these two phases. In phase 1, Actin entropy increases throughout the entire protrusive front as the cell begins to repolarize (Fig 7b panels 0 min vs 4 min). In phase 2, the entropy returns to a lower steady state pattern with elevated entropy values limited to the leading edge of the polarized cell (Fig 7b panels 17 min and 21 min). This is consistent with the notion that high Actin turnover in membrane ruffles and transversal arcs all around the cell edge of the non-polarized cell is shifted during symmetry breaking towards the wide lamellipodia and lamella regions at the new cell front [39]. This reorganization of Actin dynamics is paralleled by a reorganization of high Profilin coherence (Fig 7c). Specifically, in phase 1 we see high Profilin coherence along the arc of the cell edge that will later become the leading edge in the polarized state, on top of transversal arcs and around the nucleus. We see background levels of coherence near the retraction fibers that will later become the cell rear. Locally high Profilin coherence indicates high spatial coupling in the concentration fluctuations of Profilin-Actin complexes that are fed into Actin polymers by nucleators such as formins, which promote the growth of linear Actin structures at the future leading edge [25, 26, 37]. Importantly, it is unlikely that these distinct zones of high Profilin-Actin interactions result from artifacts in the remapping process. The size of these zones significantly

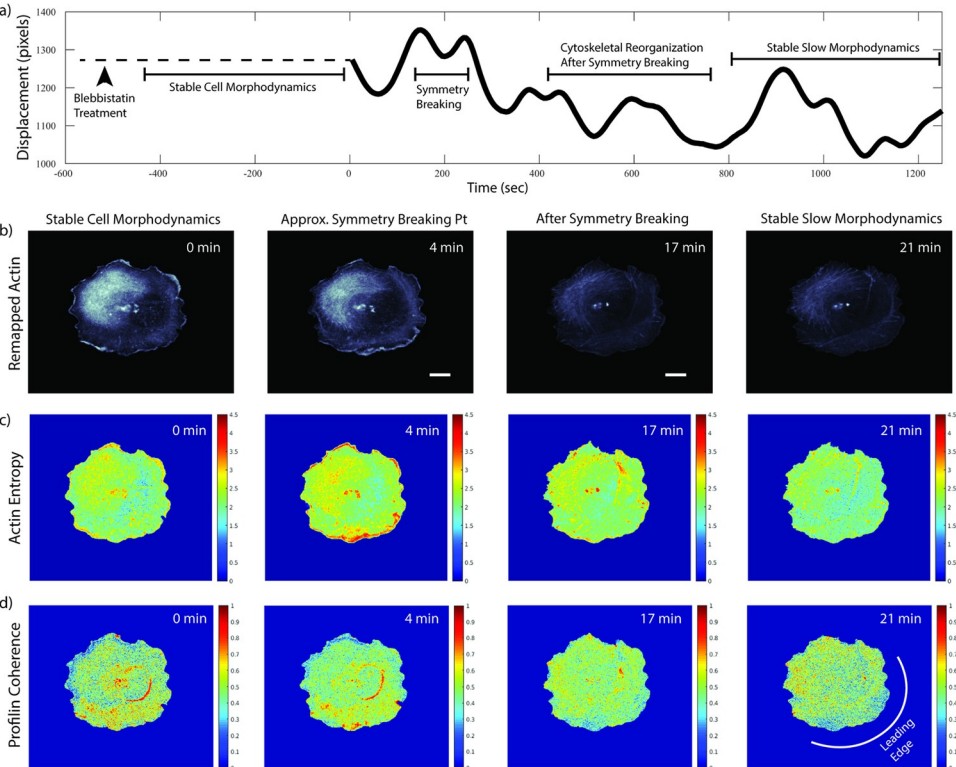

**Fig 7. Cell symmetry breaking reveals polarization dependent organization of Profilin coherence** a) Time course of a drug-induced symmetry breaking experiment, following the protocols published in [39]. 500 s prior to imaging cells seeded on glass slides are treated with 25 μM Blebbistatin. During the initial phase after drug induction the cells exhibit a stable morphology (not filmed). The remainder of the displacement time course (total cell edge displacement integrated over the cell boundary) displays distinct phases of cell morphodynamic activity, as indicated. Symmetry breaking occurs 150–250 s into the movie. b) Remapped mNG-Actin signals at four select time points. See Fig 1b for the original image sequence. The cell registration relied on Halo-CAAX as a location fiducial. c) Actin entropy computed over a rolling window of 25 time points. d) Profilin time-series coherence score computed over a rolling window of 25 time points. Both the Actin entropy and Profilin coherence reveal a shift in high values away from the cell front to the cell center and back during the symmetry breaking process showing that Profilin organization is responsive to changes in subcellular Actin polymerization. See text for further discussion. All scale bars 10 μm.

exceeds the deformation of the movie frames. Moreover, the facts that the R88E mutant and WT Profilin filmed in the same cell exhibit different dynamics along the edge (Fig 6c) and that the more stable future cell rear where we expect lower Actin turnover does not exhibit high coherence leads us to conclude that these data faithfully report effects driven by the Profilin-Actin interaction. Hence, enabled by the proposed remapping algorithm, this analysis visualizes for the first time directly in a living cell Profilin's function and organization as a facilitator of Actin assembly.

## Discussion

In this work we implemented a nonlinear cell registration framework to analyze subcellular protein dynamics in cells undergoing substantial morphological variation throughout the observation window. The key contribution of our work to the sizable literature on nonlinear image registration algorithms is the capacity to handle the high noise and small structures of interest present in cell microscopy. This was accomplished by introducing a cellular motion mask as a regularization term in the image mapping objective function and by enforcing

diffeomorphism via a sorting of the displacement field. The former causes the map estimator to bypass local minima, the latter permits the preservation of fine-grained structures during the mapping.

The primary gain this fine-grained registration framework lends to the study of cellular processes is the capacity of extracting reliable time-series at every pixel position in the cell footprint. As demonstrated in the present work, the availability of such time-series enables the further processing of image signals for information on the spatiotemporal organization of molecular processes that is inaccessible without consideration of the temporal dynamics. Specifically, we introduce the local time-series coherence and the Shannon information entropy as transforms of diffuse image signals that unveil hidden structures of spatial organization in the underlying molecular regulation. We illustrated these new features first by identifying narrow zones of coherent molecular signaling activities and then in a study of the relationship between Actin dynamics and its regulator Profilin. Fluorescently labelled Profilin generates a diffuse image signal with visually uninterpretable spatiotemporal variation. Fluorescent Actin generates a mixture of structured and amorphous image signal components. As a cell undergoes morphological changes, the structured components often display complex patterns of deformation whereas the amorphous components undergo flows that are difficult to track. Because of the proposed time-series transformations both the Profilin and Actin signals displayed spatiotemporal patterns that revealed Profilin and Actin interactions, even during a cellular symmetry breaking event that produces large scale cell shape changes. It should be noted that while the presented registration enables time-series analyses of interactions between diffuse molecular signals in a fixed cell-frame of reference, for many studies the subcellular advective motion that is eliminated with the registration is the focus of interest. In this case, users of the framework can either exploit the diffeomorphic mappings as an estimator of the advective motion components, or directly track advective movements by techniques such as quantitative fluorescent speckle microscopy and derivatives[41].

While designing this framework we expected the resulting time-series to strongly depend on the choice of the location fiducial used for the mapping estimation. We were surprised that a punctate signal (Vasp) permits the algorithm to remap Actin structures over time with an accuracy that is comparable to the mapping accuracy supported by a location fiducial derived from a diffuse signal like Profilin or a blurred, lowpass-filtered version of the Actin signal itself. This is likely because adhesion proteins like Vasp have a faint but implicitly traceable diffuse component that constrains the estimation of the diffeomorphic mapping between frames similarly to the Profilin and blurred Actin signals. Moreover, cytoskeleton structures are highly coordinated in healthy cells and the cytoplasm is a dense compartment meaning that subcellular molecular flows in general are coupled. Hence, there is some degree of tolerance in choosing a location fiducial.

Our framework opens the door to analyses of subcellular signals with dynamics that occur on the same or slower timescale as cell morphological changes. For processes much faster than cell morphological changes (e.g. cell electrical potentials and calcium signaling), microscopy has produced predictive quantitative models for *in vivo* signaling behavior and outcomes because changes in cell morphology could be ignored. However, many molecular processes occur in concert with cell morphodynamics. While previous work has mostly relied on visually-guided analyses of these processes in select cell regions, the presented framework now supports an unbiased analysis across the entire cell.

The current software implementation is restricted to 2D time lapse image sequences of cells cultured on flat substrates, although Eqs (1), (3) and (4) are generalizable to 3D data. For practical purposes this means that the software can only be run on movies of cells whose footprint is kept in focus over the entire recording. The biological artifacts associated with cell culture

on flat and often stiff substrates are widely discussed and intense developments are ongoing in microscopy design and image analysis to circumvent these caveats [42, 43]. Nonetheless, a significant portion of imaging-based cell biological investigations still relies on 2D live cell microscopy. These studies may readily benefit from the presented work.

Related to the limitation to 2D imaging are potential sources of inaccuracy that need to be considered when applying the accompanying codes: As introduced with the choice of Thirion's demons and expanded regularization, the cell registration algorithm assumes that the motion of the location fiducials is i) fully visible throughout the movie and ii) restricted to the directions parallel to the substrate. While these assumptions are mostly valid for adherent cells, weakly adherent cells with substantial extension in the direction perpendicular to the substrate may not be analyzable with the proposed framework. Dependent on the dynamics of the location fiducial and signal of interest this caveat can be remedied partially by rapid 3D imaging of the full cellular volume by light-sheet microscopy [44], followed by projection of the 3D signal to 2D. Even for adherent cells uncertainty arises in the perinuclear region and/or in vertical membrane ruffles, where both the location fiducial and signal of interest contain significant out-of-focus intensity contributions when imaged by wide-field microscopy and suffer from non-detectable axial movement, regardless of the 2D microscopy mode. Both imaging limitations cause a violation of the assumption of intensity conservation underlying Thirion's demons. However, one of the strengths of the proposed regularization approach is the implicit propagation of geometric information from regions of the cell with full in-focus data and largely lateral deformation to regions with geometric information loss. The inclusion of the cell boundary provides particularly stabilizing support to the estimation of diffeomorphic maps under conditions of incomplete location fiducials (see also Fig 2c). Thus, for many practical applications the proposed algorithm may prove robust enough.

The second assumption requiring consideration with applications of the proposed framework is the geometric coupling of location fiducial and signal of interest. For example, cell registration of a cytoplasmic molecule with a location fiducial that is localized in the plasma membrane or even in close contact with the substrate will produce poor results. The proposed framework is best suited for registration of dynamic and relatively diffuse patterns of molecular distributions and activities using dense cytoskeleton structures as location fiducials. Cytoskeleton structures tend to follow the morphodynamic behavior of the cell at large and many of the molecular components in cytoplasm and even in the plasma membrane tend to directly or indirectly, e.g. via cytoskeleton-driven fluid flows, interact with these polymer assemblies. In the present study we illustrate this performance by mapping out for the first time the dynamic organization of the cytoplasmic molecule Profilin. In summary, despite these limitations we anticipate a wide range of applications that will benefit from the potential of the proposed framework to spatially align patterns of molecular behaviors in a frame of reference that compensates for cell morphodynamic activity.

## Materials and methods

### Plasmids

pSpCas9(BB)-2A-GFP (PX458) and pX335-U6-Chimeric_BB-CBh-hSpCas9n(D10A) were gifts from Dr. Feng Zhang (Addgene plasmid #48138 and #42335, respectively). Gene-targeting single guide RNAs (sgRNAs) were designed using the online program CRISPor (http://crispor.tefor.net) [45]. pmCherry-C1 was from Clontech. The self-cleaving vector pMA-tial1 was from Dr. Tilmann Bürckstümmer [46]. pBlueScript II SK(+) was from Agilent. The primers to clone the Profilin-1 (PFN1)-targeting sgRNA pair used for knockouts were 5'- CAC CGTCGATGTAGGCGTTCCACC-3' and 5'-AAACGGTGGAACGCCTACATCGAC-3' and

were cloned into PX458. PFN1-targeting sgRNA pairs for the knock-in were 5'- CACCGGC TGCTACTGGGGCTGCTCTCGG -3', 5'- AAACCCGAGAGCAGCCCCAGTAGCAGCC -3', 5'- CACCGCGCCTACATCGACAACCTCATGG -3' and 5'- AAACCCATGAGGTTGTCG ATGTAGGCGC -3', all cloned into PX335. The self-cleaving donor vector containing the Blasticidine selection cassette was previously described [47, 48]. The N-terminal SNAP-tag knock-in donor was flanked with homologous arms targeting the first exon of PFN1 (BAC library: CH17-25E2) including a 13 amino acid linker (SGRTQISSSSFES) in between SNAP-tag and PFN1, a configuration well-characterized to preserve all functional properties of Profi-lin-1 [49, 50], and cloned into pBlueScript II SK(+). pLVXCMV100-mNeonGreen-18-Actin and pLVXCMV100-Halo-21-VASP were previously described [29]. pLVXCMV100-Halo-CAAX was generated by Gibson Assembly (NEB) using HaloTag sequence as a template and the following primers: 5'-attaactagtgccaccatggcagaaatcggtactggctttcc-3', 5'-ttacataattaca-cactttgtctttgacttctttttcttcttttttaccatctttgctcatctttttctttatgGCCGGAAATCTCGAGCGTCGAC-3' and 5'-attacgcgtTTACATAATTACACACTTTGTCTTTGACTTCTTTTTCTTC-3'.

N-terminally tagged EGFP-C-Profilin-10 and mApple-C-Profilin-10, both harboring mouse Profilin-1 and a 10 amino acid linker (SGLRSRAQAS) were gifts from Michael David-son (Addgene plasmids #56438 and #54940). The Actin-binding deficient Profilin-1 mutant R88E [51] was generated by mutating the Arginine at position 88 on Profilin to Glutamic acid (R88E) using primers 5'-gagACCAAGAGCACCGGAGGAGCCCC-3' and 5'-AAGATCCAT TGTAAATTCCCCGTCTTGCAGCAGTG-3' and PfuUltra II Fusion High-fidelity DNA poly-merase (Agilent). EGFP-Profilin and mApple-Profilin were subsequently subcloned into the SpeI/MluI sites of the lentiviral vector with attenuated CMV promoter, pLVXCMV100 [52], using primers 5'- ttaactagtgCCACCATGGTGAGCAAGGGCGAG -3', 5'- ttaactagtgccaccA TGGTGAGCAAGGGCGAGGAGAATAACATGG -3' and 5'- aaacgcgtTCAGTACTGGGA ACGCCGCAGGTGAGA -3'. All newly generated constructs were sequence verified.

## Antibodies

Mouse monoclonal anti-Profilin-1 (Santa Cruz; B-10; sc-137235), mouse monoclonal anti-vin-culin (Sigma; V9264) and mouse monoclonal anti-Actin (Sigma; AC-15; A1978) antibodies.

## Cell lines

Human Osteosarcoma U2OS cells were cultured in DMEM media supplemented with 10% fetal bovine serum (Sigma; F0926-500ML) in a humified incubator at 37°C and 5% $CO_2$. All cells were tested for mycoplasma using a PCR-based Genlantis Mycoscope Detection Kit (MY01100).

Lentiviral particles were generated using the packaging vectors psPAX2 and pMD2.G (Addgene plasmids #12260 and #12259). Infected cells were bulk sorted using FACS.

Profilin-1 (PFN1) knockout U2OS cells were generated by co-transfecting the PFN1-target-ing sgRNA supplemented with the self-cleaving Blasticidine selection cassette. Genome-edited cells were selected using 5 μg/ml Blasticidine S selection (Thermo) and isolated using 8 mm colony cylinders (Sigma). Knockouts were verified using western blot with mouse anti-Profi-lin-1 antibodies. Profilin KO clone #2 were used for subsequent experiments (S2 Fig).

Endogenously SNAP-tagged Profilin-1 cells were generated by co-transfecting the two pairs of PFN1-targeting sgRNAs with Cas9 nickase supplemented with the donor vector. Genome-edited cells were labeled with SNAP-Cell Oregon Green (S9104; NEB) and single cell sorted into 96 wells plated coated with attachment factor (S006100; Gibco). Successful genome-edited cells were validated using western blotting.

mCherry was expressed in U2OS cells expressing mNeonGreen-Actin and Halo-CAAX by transient transfection with polyethylenimine and 1 μg of pmCherry-C1 one day prior to imaging. U2OS Profilin-1 KO U2OS cells were rescued with wild-type EGFP-Profilin and Actin-binding deficient mApple-Profilin mutant (R88E) using a lentiviral construct driven by a truncated CMV promoter as discussed above. Infected cells were bulk sorted using FACS.

## Live cell imaging

Cells were counted using Cellometer Auto 1000 Bright Field Cell Counter (Nexcelom) and 100.000 cells were seeded on 10 μg/ml fibronectin-coated glass bottom 35 mm. Halo-CAAX was labeled using Janelia Fluor 549 HaloTag ligand (200–400 nM; GA1110; Promega) and endogenous SNAP-Profilin-1 were labeled using SNAP-Cell 647-SiR (500 nM– 1 μM; S9102S; NEB), 30 minutes prior to imaging. Cells were imaged in phenol-red free DMEM supplemented with 20 mM HEPES pH 7.4 on a climate-controlled (maintained at 37˚C), fully motorized Nikon Ti-Eclipse inverted microscope equipped with Perfect Focus System, an Andor Diskovery illuminator coupled to a Yokogawa CSU-XI confocal spinning disk head with 100 nm pinholes, and a 60x (1.49 NA) APO TIRF objective (Nikon) with an additional 1.8x tube lens, yielding a final magnification of 108x (Andor Technology). Images were recorded at 5 s or 10 s per frame using a scientific CMOS camera with 6.5-μm pixel size (pco.edge).

## Blebbistatin treatment to induce symmetry breaking

Cells were treated with 25 μM myosin II inhibitor blebbistatin in DMEM for 5 min prior to imaging as per [39]. Drug treatment was offset to allow for a complete 20 min imaging run per dish under the expectation that symmetry breaking would occur at approx. the 10 min mark. Fields were selected manually to center a single individual cell and cells were selected for a flat rounded morphology. After imaging, the movies were examined for spontaneous symmetry breaking near the midpoint. We utilized the first 1/3 of the movie as before symmetry breaking samples and the last 1/3 of the movies as post symmetry breaking samples.

## Remapping pipeline parameters

For all published results we set $\alpha = 1$ for a maximum step size of 1 pixel. For sequentially transformed movies for time-series analysis we set $n = 100$ iterations per time step and $K_{diff} = 1.5$ pixels. For transforms between the first and last image frames we set $n = 2000$ iterations. For ½ deformation accuracy we set $n = 200$ iterations.

## Coherence analysis

For every subcellular location we sampled a 3x3 matrix of 9 time-series centered on the target pixel. We calculated the correlation coefficient for all 36 = non-identical time-series pairs, where the correlation coefficient $\rho(A, B)$ between time-series $A$ and $B$ is:

$$\rho(\boldsymbol{A}, \boldsymbol{B}) = \frac{1}{N-1} \sum_{i=1}^{N} \left( \frac{A_i - \mu_A}{\sigma_A} \right) \left( \frac{B_i - \mu_B}{\sigma_B} \right)$$

$\mu_A$ and $\sigma_A$ are the mean and standard deviation of $\boldsymbol{A}$, respectively, $\mu_B$ and $\sigma_B$ are the mean and standard deviation of $\boldsymbol{B}$, and $N$ is the number of time points in the series $\boldsymbol{A}$ and $\boldsymbol{B}$. The coherence is the mean of the resulting coefficients. We restricted the coherence analysis to pixels whose 3x3 neighborhood fell completely inside the cell mask in the reference frame. We calculated coherence values based on time-series over the entire movie or in moving windows of 20 frames (1/10 length of typical movie) to analyze coherence change during symmetry breaking.

### Actin entropy calculation

The mNG-actin channel visualizes a mixture of local monomers, branched Actin, and linear bundles. To assess the local concentration change in all of these entities we calculated the Shannon information entropy Actin in moving windows of 20 frames (1/10 the length of a typical movie). For every Actin intensity series $X$ the entropy $H(X)$ is defined as:

$$H(X) = \sum_{i=1}^{N} P(X_i) \log_2 P(X_i)$$

Where $X_i$ is a value in series $X$ and $P(X_i)$ is the probability of drawing $X_i$ from series $X$. Actin

### Windowing parameterization

Cell edge-tracking probing windows were defined with the 'Windowing-Protrusion' software package available on our Github site (https://github.com/DanuserLab). The underlying algorithm is published in [1, 8]. We used the option 'Constant number' as a method of propagating the windows to the next time frame. The size of the probing windows was set to 600×600 nm$^2$, comparable to previous work examining Actin dynamics in U2OS cells [29]. Over time and between window layers, the width is variable to follow the cell edge movement. The depth of the windows remains fixed.

### Supporting information

**S1 Fig. Impact of location fiducial on remapping accuracy (related to Fig 3).** a) Reference frame images of the three location fiducials, lowpass-filtered Actin, Profilin, and CAAX in the same U2OS cell displayed in Fig 1b) To-target transformation and half-distance transformation accuracies (sum of squared distance (SSD) between target and remapped images) computed for the full mNG-Actin signal of interest (see Fig 1) using different location fiducials as indicated. The SSD between untransformed moving and target frames is computed as a baseline. Box plots illustrate 25$^{th}$, 50$^{th}$, and 75$^{th}$ percentile of n = 91 moving/target frame pairs pooled from m = 4 movies. Whiskers indicate the 5$^{th}$ and 95$^{th}$ percentile. P-values are calculated by one-way ANOVA testing.
(PDF)

**S2 Fig. Verification of Profilin knockout clones.** Profilin knockout was verified using western blotting using mouse monoclonal anti-Profilin-1 antibodies. Vinculin and Actin provided as loading control. See materials and methods for antibody source.
(PDF)

**S1 Movie. Cell registration of an Actin signal of interest based on a CAAX location fiducial (related to Fig 1).** Cell: U2OS SNAP-CRISPR-Profilin, mNG-Actin, Halo-CAAX treated with 25 μM Blebbistatin. Top left: CAAX original movie. Top right: CAAX remapped movie. Bottom left: Actin original movie. Bottom right: Actin remapped movie. Acquisition rate: 1 frame/5s. Replay rate: 21 frames/s
(AVI)

**S2 Movie. Registration of cell edge before and after remapping (related to Fig 1).** Cell: U2OS SNAP-CRISPR-Profilin, mNG-Actin, Halo-CAAX treated with 25 μM Blebbistatin. Left: original movie. Right: remapped movie. Color scale: red-current time point edge position -> blue-past time point edge position. Acquisition rate: 1 frame/5s. Replay rate: 21 frames/s
(AVI)

**S3 Movie. Windowing of transverse arcs (related to Fig 3e and 3f).** Cell: U2OS SNAP-CRISPR-Profilin, mNG-Actin, Halo-CAAX treated with 25 μM Blebbistatin. Left: full cell view. Right: zoomed in view. Color: red-subcellular windows, pink-transverse arc detection, black lines-window boundaries. Acquisition rate: 1 frame/5s. Replay rate: 7 frames/s
(AVI)

**S4 Movie. Comparison of Profilin signal and Profilin Coherence (related to Fig 5b).** Cell: U2OS SNAP-CRISPR-Profilin, mNG-Actin, Halo-CAAX treated with 25 μM Blebbistatin. Left: Profilin signal. Right: Profilin coherence in 25 time point moving windows. Acquisition rate: 1 frame/10s. Replay rate: 7 frames/10s
(AVI)

**S5 Movie. Comparison of WT-Profilin and R88E-Profilin coherence in same cell (related to Fig 5c).** Cell: U2OS EGFP-Profilin[WT], mCherry-Profilin[R88E], SNAP-Actin. Left: WT-Profilin coherence in 25 time point moving windows. Right: R88E-Profilin coherence in 25 time point moving windows. Acquisition rate: 1 frame/10s. Replay rate: 7 frames/10s
(AVI)

**S6 Movie. Comparison of Actin signal and Actin entropy (related to Fig 6).** Cell: U2OS SNAP-CRISPR-Profilin, mNG-Actin, Halo-CAAX treated with 25 μM Blebbistatin. Left: Actin signal. Right: Actin entropy. Acquisition rate: 1 frame/5s. Replay rate: 7 frames/5s
(AVI)

**S7 Movie. Profilin coherence before and after symmetry breaking (related to Fig 7).** Cell: U2OS SNAP-CRISPR-Profilin, mNG-Actin, Halo-CAAX treated with 25 μM Blebbistatin. Left column: before symmetry breaking. Right column: after symmetry breaking. Top row: original Actin signal. Mid row: remapped Actin signal. Bottom row: Profilin coherence in 25 time point moving windows. Acquisition rate: 1 frame/5s. Replay rate: 7 frames/5s
(AVI)

## Acknowledgments

We thank Dr. Dick McIntosh (University of Colorado, Boulder, CO) for providing the U2OS osteosarcoma cells, Dr. Tilmann Bürckstümmer and all Addgene depositors for sharing reagents and Dr. Dana Reed (UT Southwestern Medical Center) for her logistical support and laboratory management. We thank our collaborators Drs Hahn and Marston (University of North Carolina, Chapel Hill) for re-sharing previously published movies of GTPase and GEF signaling activity. We thank Roshan Ravishankar and Qiongjing (Jenny) Zou for preparation of the code distribution on Github. We also thank the UT Southwestern BioHPC facility for providing high-performance computing systems.

## Author Contributions

**Conceptualization:** Xuexia Jiang, Gaudenz Danuser.

**Data curation:** Xuexia Jiang, Tadamoto Isogai.

**Formal analysis:** Xuexia Jiang.

**Funding acquisition:** Gaudenz Danuser.

**Investigation:** Xuexia Jiang.

**Methodology:** Xuexia Jiang, Tadamoto Isogai, Gaudenz Danuser.

**Project administration:** Gaudenz Danuser.

**Resources:** Joseph Chi.

**Software:** Xuexia Jiang.

**Supervision:** Gaudenz Danuser.

**Validation:** Xuexia Jiang.

**Visualization:** Xuexia Jiang, Gaudenz Danuser.

**Writing – original draft:** Xuexia Jiang.

**Writing – review & editing:** Xuexia Jiang, Tadamoto Isogai, Gaudenz Danuser.

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
