## [Decision Letter · Decision Letter 0]

27 Jan 2022

Dear Dr. Danuser,

Thank you very much for submitting your manuscript "Fine-grained, nonlinear image registration of live cell movies reveals spatiotemporal organization of diffuse molecular processes" for consideration at PLOS Computational Biology.

As with all papers reviewed by the journal, your manuscript was reviewed by members of the editorial board and by several independent reviewers. In light of the reviews (below this email), we would like to invite the resubmission of a significantly-revised version that takes into account the reviewers' comments.

We cannot make any decision about publication until we have seen the revised manuscript and your response to the reviewers' comments. Your revised manuscript is also likely to be sent to reviewers for further evaluation.

Sincerely,

Melissa L. Kemp, Ph.D.

Associate Editor

PLOS Computational Biology

Jason Haugh

Deputy Editor

PLOS Computational Biology

Reviewer's Responses to Questions

**Comments to the Authors:**

Reviewer #1: This article addresses an important yet often underappreciated problem of cell mapping for fluorescent images in cases where no simple ground truth position marker is available, and where the focus is on tracking of diffuse protein dynamics. While pixel scale dynamics is often notable by eye in these movies, quantitative analysis and comparison across experiments is difficult due to changes in cell shape. The method introduced by the authors represents an important advance to the field. However, the constraints and limitations of the new method are not described in the manuscript, and the improvements over other approaches appear exaggerated.

While the authors describe the assumptions for the pipeline in lines 117 onward, they state assumptions that are never completely fulfilled in particular the assumption that the cell sits flat in 2D. There are two specific concerns. First, as described, the images appear to be traditional fluorescence images of cells that are fully in the field of view (that is never stated or explained, but should be). Thus changes in fluorescence intensity can also indicate changes in the local thickness of the cell. Does this mean that the cell registration projects changes in thickness as lateral displacements of the cell? How accurately does a 2D optical flow model capture the 3D flow inside the cell that is being captured? Would this approach also be applicable to confocal images?

The second concern is that such a remapping may not be suitable for proteins that are bound to the surface directly or indirectly - the laboratory reference frame would appear to be more suitable for analysis of such proteins.

Additional comments:

The authors describe their approach as image registration, which is more commonly used to compensate for fluctuations in the imaging field of view rather than compensating for cell movement and deformation. Since traditional image registration is also an important aspect of any data analysis it will be important for the authors to clearly differentiate their effort on cell registration from image registration work.

The authors describe the prevalent analysis approach to such data as “anecdotal” and “superficial” which is not an accurate representation of the quantitative prior work in this area, including work by the authors. I recommend a more nuanced description of prior work in this area.

Since optical flow is the key enabling technique, an expanded discussion with references of the following statement would be important – line 72 “However, because of the aperture problem and image noise, Optical Flow computation is ill-posed and thus requires domain-specific regularization techniques.”

Coherence is a 3 x 3 region if I understand the explanation – This will introduce asymmetries. Did the authors verify that the asymmetry of this metric does not lead to artifacts when it is applied along boundaries? In other words, Does coherence depend on the angle of the cell boundary relative to the image axes?

Line 93 “appreciate” - the scientific meaning of this term is unclear.

The abstract describes a study of “signaling” as separate from “protein distributions” but the experiments shown only focus on fluorescently labeled proteins. Please clarify

Reviewer #2: An interesting article well satisfying criteria for publication in PLOS Comp Biol. It describes an improved method of elastic registration based on optical flow with the help of the motion mask as a location fiducial while maintaining diffeomorphism using sorting-based regularization.

Authors show the efficiency of the algorithm with the help of ‘to-target transform’ and ‘half-distance transform’ applied to a sequence of U2OS cells and convincingly demonstrate its applicability to showing dynamics of Profilin and Actin in these cells.

I would have the following comments:

In p. 11, lines 201-204 you mentioned panels i-iv, but there are no such panels in Fig. 2.

In the same page, l. 209, you mentioned „2 distant time points“. Please, use a word for the number and I propose to specify this time distance, also l. 214.

In Figure 4 there should be probably „Vol. Marker mCherry Coherence“ as a title for the top-right picture. Possibly, the Intensity and Coherence pictures in the last row were swapped (R88E), since the dispersions in the graph below appear not to correspond to visualized pictures.

In p. 38, l. 767: In my opinion, “the band 0 – 3.6 µm“ does not correspond to the previously stated “1.2 µm from the cell edge”. Please, complete also the description of the vertical axes of the graphs in e).

I had trouble joining the description both in the text (p. 19) and the legend with Figure 6. It is not clear to me what is the cell front and from the small pictures what exactly happens. All the visualized ceIls are practicaly the same when printed on paper. I propose to apply here, e.g., zoomed-in insets, arrows in the pictures that would help readers to understand visualized and described processes, though it is true that Movie 8 helps with this.

It appears to me that Movie 1 and Movie 2 – top row are identical.

Please, correct these small typos:

p. 8, l. 134: structures

p. 10, l. 191: put the reference before the dot

p. 17, l. 344: selected

Reviewer #3: In this manuscript, the authors present a method for registering cellular images adapting from optical flow-based approaches with an updated regularization component that includes the constrain of cell boundaries location. Overall the proposed methods look quite promising in realigning/transforming the cellular images to a reference image of cell at the reference time points. However, the utility of the proposed method in analyzing and interpreting cell dynamics and biological applications was not very clear and the study of the profillin dynamics by applygin the proposed registered images is very descriptive and lack of stasistical significance. Here are my main concerns:

1) Though the authors show that using their updated method, they can re-map the cell images toward the reference images with great similarity, it is not clear the biological representaion of these transformed images referenced at the target images as the ground truth is difficult to establish. The utility of the proposed method to study system dynamics should be further validated in biological dynamics systems with a known/expected outcome.

2) The authors applied their registration method to study the profilin dynamics where they used the CAAX membrane marker as their fiducial images. However, The author only evaluated the performance of using profilin, smoothed actin, and VASP as fiducial points in figure 3, and the performance of CAAX as fiducial points was not evaluated. Hence, it's unclear how well the registration performs based on CAAX images to study the profilin dynamics.

3) “..we hypothesized that the same principle of spatial coherence in dynamic behavior could be applied to map out the subcellular organization of protein dynamics…” the spatial coherence appears to be a key characteristic that authors measured to assess the dynamics of profillin but what this characteristic represents is unclear. “…as expected, coherence of a diffuse signal is high across the entire cell footprint….”. It's not clear what to expect here. The authors should provide a more detailed explanation of the meaning of spatial coherence. In the method section for coherence measurement, the author mentioned that the coherence is measured using a 3x3 matrix of 9-time series centered on the target pixel and measure correlation coefficient for all 72 =non-identical time-series pairs from 9-time points images. It's not clear how 3x3 and 9-time points is arrived and it seems that there should be only 36 non-identical pairs.

4) Its general lacking of statistical description/significance in the profillin dynamic study using the proposed algorithm (figure 4 to figure 6). In Figures 4 and 6, it seems there is only one cell is measured in each comparing condition, and hence the statistical significance of the observed phenomenon is unclear. In Figure 5, it seems there are multiple measurements (Figure 5e), but there is no description of sample size and the number of independent replicates. There is also no statistical test to evaluate the significance of differences. Overall, more cells from biological replicates should be measured to further validate the finding.

5) The figure caption for figure 1D is not clear and does not reflect the statement, “The mapping into a rigid frame of reference permits straightforward sampling of times series”.

6) There is no statistical information provided for figure 3D. What do the measurements in the box plot represent?

7) Figure 4A, the figure title label seems to be wrong

**Have the authors made all data and (if applicable) computational code underlying the findings in their manuscript fully available?**

Reviewer #1: Yes

Reviewer #2: Yes

Reviewer #3: Yes

PLOS authors have the option to publish the peer review history of their article (what does this mean?). If published, this will include your full peer review and any attached files.

Reviewer #1: No

Reviewer #2: No

Reviewer #3: No
---

## [Decision Letter · Decision Letter 1]

16 Sep 2022

Dear Dr. Danuser,

Thank you very much for submitting your manuscript "Fine-grained, nonlinear registration of live cell movies reveals spatiotemporal organization of diffuse molecular processes" for consideration at PLOS Computational Biology. As with all papers reviewed by the journal, your manuscript was reviewed by members of the editorial board and by several independent reviewers. The reviewers appreciated the attention to an important topic. Based on the reviews, the journal will accept this manuscript for publication, providing that you modify the manuscript according to the minor edits requested by Reviewer 1.

Sincerely,

Melissa L. Kemp, Ph.D.

Academic Editor

PLOS Computational Biology

Jason Haugh

Section Editor

PLOS Computational Biology

[LINK]

Reviewer's Responses to Questions

**Comments to the Authors:**

Reviewer #1: The revised manuscript addresses all concerns of this reviewer and provides a clear introduction to the valuable new tool developed by the authors. I would like to request the authors review one particular wording choice and correct several minor errors.

The manuscript could be clearer on the nature of the corrections introduced by the authors. The goal is stated in the introduction – highlighting molecular interactions. As I understand, the authors wish to remove advection effects from the motion (which would impact all interacting molecules in the same way) and only retain reactions, active transport e.g. via motors or passive diffusion. It should be clarified that advective subcellular movements can also be of importance.

The following minor errors should be addressed based on the redlined copy:

In the abstract the wording is incorrect: “in function of”

Line 72 there is a word missing “the method enables the registration the cell outline”

Line 223 “5 s per frame to 50 s per frame.” This could be misunderstood as a change in the exposure time of the camera.

Line 236 actin appears twice.

Line 248 onward – random words are not crossed out.

Line 286: there seems to be a mistake in the wording – is the new method still only useful for “the entire cell perimeter” or does the remapping also work for the interior of the cell?

Reviewer #3: This reviewer has no further question.

**Have the authors made all data and (if applicable) computational code underlying the findings in their manuscript fully available?**

Reviewer #1: Yes

Reviewer #3: Yes

PLOS authors have the option to publish the peer review history of their article (what does this mean?). If published, this will include your full peer review and any attached files.

Reviewer #1: No

Reviewer #3: No

Figure Files:

Data Requirements:

Reproducibility:

References:

---

## [Editor Report · Decision Letter 2]

28 Nov 2022

Dear Dr. Danuser,

We are pleased to inform you that your manuscript 'Fine-grained, nonlinear registration of live cell movies reveals spatiotemporal organization of diffuse molecular processes' has been provisionally accepted for publication in PLOS Computational Biology.

Best regards,

Melissa L. Kemp, Ph.D.

Academic Editor

PLOS Computational Biology

Jason Haugh

Section Editor

PLOS Computational Biology

---

## [Editor Report · Acceptance letter]

27 Dec 2022

PCOMPBIOL-D-21-02096R2 

Fine-grained, nonlinear registration of live cell movies reveals spatiotemporal organization of diffuse molecular processes

Dear Dr Danuser,

I am pleased to inform you that your manuscript has been formally accepted for publication in PLOS Computational Biology. Your manuscript is now with our production department and you will be notified of the publication date in due course.

With kind regards,

Bernadett Koltai
